# Testing the Ricardian Equivalence Theorem: Time Series Evidence from Turkey

Ahmet Salih İkiz 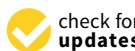

Faculty of Economics and Administrative Sciences, Muğla Sıtkı Koçman University, Muğla 48000, Turkey; ahmet@mu.edu.tr

**Abstract:** Two of the most common measures adopted by the government to stimulate the economy are increasing government borrowings and implementing tax cuts. These tax cuts are financed through increased debt. According to the Ricardian equivalence theory, the consumers will not change their current spending when they anticipate a tax increase in the future. In order to pay high taxes in the future, the government should increase its present savings. However, the extent of applicability of Ricardian equivalence could vary across nations. In this context, the present study explores the long-running relationship between domestic borrowing and private savings in Turkey. For this purpose, the researcher collected the data for key variables, gross domestic savings, and government debt, for the period of 1980–2017. The researcher used unit root, cointegration, VECM, and the Granger causality test to examine the relationships among the variables. Apart from this, ARDL regression was used in order to examine the long-term relationships among the variables. The empirical results indicate that there is presence of bidirectional causality, indicating that Ricardian equivalence is applicable in the economy. Households display a rational behavior by increasing their savings during the periods in which high government expenditure is incurred.

**Keywords:** Turkey; government expenditure; private savings; Ricardian equivalence; taxes; consumption

## 1. Introduction

### 1.1. The Impact of the Rise in Government Debt

Economic liberalism in government functions was deserted with the increasing importance of social state policies in post war period. The minimal state functions used to be education, security, and justice, for which governments need few financial resources for a sustainable budget. Transformation of those functions to a broader level with a high burden of expenditures led to high financial resource requirements for public spending. In turn, government debt has risen to extraordinary levels.

Management of government debt is a worldwide issue being looked by the legislatures of many developing and developed nations (Maslov 2015). The macroeconomic impacts of government debt have developed the need for restoration of the academic and policy arguments on the effect of developing debt levels on economic growth. These arguments arise due to inefficient management of government debt which can negatively impact public or private investment and the savings of the economy (Odim et al. 2014). On the contrary, if borrowing has been allotted effectively, an economy can achieve profits, as debt financing of public spending can make positive contributions to gainful investment and indeed to economic development (Slav'yuk and Slaviuk 2018).

A rising level of government debt can create a high level of risk for investments and even create difficulties for investors in the lending markets. In addition to this, it raises the issue of adequacy of the fiscal policy formulated by the government (Butkus and Seputiene 2018). Further, it also increases

the level of fiscal deficit, which in turn leads to a rise in inflation in long term. This rise in level of inflation has a major impact on household behavior in the form of reduction in consumer spending. This reduction in the consumer spending eventually leads to a rise in private savings (Sains 2016).

*1.2. Ricardian Equivalence Theory*

The Ricardian equivalence theorem (RET) plays an important role in macroeconomic theory. The RET suggests that fiscal stimuli which are defined in terms of deficit-financed public spending hikes or tax cuts will lead to a crowding out of private consumption, thereby decreasing the effectiveness of fiscal policy in boosting economic activity (Hayo and Neumeier 2017). It stipulates that a person's consumption is determined by the lifetime present value of his after-tax income. Therefore, the Ricardian equivalence says a government cannot stimulate consumer spending, since people assume that whatever is gained in the present will be offset by higher taxes due in the future. Thus, the underlying idea behind Ricardo's theory is that no matter how a government chooses to increase spending, whether by debt financing or tax financing, the outcome is the same and demand remains unchanged (Hayo and Neumeier 2017).

Barro's treatment of Ricardian comparability has received acceptance in the setting of national government finance. The modern Ricardian equivalence hypothesis in this manner probably abstains from whether increased government spending harms an economy's development (Ahiakpor 2013). Guided by the Ricardian equivalence model, economists have would in general affirm that consumers ought to be indifferent between bond financed and tax financed investments or what is called "debt illusion" (Banzhaf and Oates 2012). Under the assumptions of an infinite horizon, non-distortionary taxes, an absence of liquidity constraints, farsighted people, and an ideal capital market, a present tax cut will not increase consumption, since non-myopic people will see this arrangement as an increase of taxes later on (Drakos 2001).

*1.3. Economic Indicators Affected by RET*

RET contends that for a given way of government spending, a deficit financed cut in current taxes prompts higher future expenses that have a similar present value as the underlying cut (Hayo and Neumeier 2017). If the Ricardian equivalence is completely true, then at that point any increase in government expenditure would prompt a related decline in consumption expenditure, as households' units would save more because they expect the future risk. The net impact on aggregate demand at that point is zero and fiscal policy is completely incapable of achieving high levels of economic growth. Ricardian equivalence in this manner infers that the financing scheme of government expenditures is superfluous. In any case, the adjustment in the change in the level of government expenditure is applicable (Choi and Holmes 2014). Likewise, this impact can vary a lot from nation to nation, and over the short-run and the long-run. There have been studies on direct trials of Ricardian equivalence in developed nations; be that as it may, such tests are not done in developing nations so regularly. Ricardian equivalence suggests that financial switching among debt and taxes will not have any impact on macroeconomic factors, such as private consumption, interest rates, and the current accounts. Regardless of huge quantitative confirmations on the implications of Ricardian equivalence for macroeconomic factors, the general outcomes are uncertain (Lindsey 2016).

## 2. Research Aim and Objectives

There are several factors in RET. For example, the permanent income hypothesis is model in which an optimized government chooses a stream of public consumption expenditures over time to maximize discounted social welfare, constrained only by its permanent income (Park 1997). Additionally, Ramsey searched the motives behind saving in a country in his study (Ramsey 1928).

The aim of the present study was to explore the applicability of the Ricardian equivalence theory in Turkey. Therefore, I attempted to discover the extent of the relationship between domestic government borrowings and private sector savings for the Turkish economy for the period of 1980–2017—the

outcomes of an increase in internal public debt and the impacts of government borrowings on household savings and consumption decisions.

## 3. Literature Review

Before moving further, it would be better to list the main assumptions of Ricardian equivalence theorem.

1. Income life-cycle hypothesis: Consumers wish to smooth their consumption over the course of their lives. Thus, if consumers anticipate a rise in taxes in the future, they will save their current tax cuts to be able to pay future tax rises.
2. Rational expectations on behalf of consumers. Consumers respond to tax cuts by realizing that they will probably mean future taxes have to rise.
3. Perfect capital markets: Households can borrow to finance consumer spending if needed.
4. Intergenerational altruism: Tax cuts for the present generation may imply tax rises for future generations. Therefore, it is assumed that an altruistic parent will respond to current tax cuts by trying to give more wealth to their children so they can pay the more increased tax in the future.

### 3.1. The Inter-Temporal Government Budget Constraint

The intertemporal budget constraint showcases the government choices between the fiscal and monetary measures, and also to decide on how much to spend as public expenditure and how much to save on future prospects of growth (Elliott and Kearney 1988). Keynesian work missed out on the impact of government budget on allocation and outcomes, which was pointed out by Barro in 1974 while reviewing on the works of Ricardo (Leeper and Nason 2008).

In a similar context, the study conducted by Buiter (2002) highlighted that the intertemporal budget constraint is related to high levels of government debt when it runs a high future surplus in term of present value and it tends to be produced through changes in taxes, government expenditure, or seigniorage. The study of Doppelhofer (1994), provided a proof to reject that the taxes can act as a way to balance the budget deficit and found it unfit to reject the theory of intertemporal budget balance (Kwang Jeong 2014); the study points out that around 65–70% of budget deficit is because of high government spending, around 50–65% of budget deficit is due decline in taxes that have been eliminated with by step-wise decreases in government spending, and the rest is eliminated by step wise increased in tax revenue. Darrat 2006 proposed that ideal strategy to combat the budget deficit issue is to raise taxes. The empirical literature above it gives us blended outcomes on the intertemporal relationship between government expenditure and taxes. The studies indicate a use of diverse methodologies depending upon various social, political, and economic factors; hence, it cannot be pointed out which approach is the best fit.

The study of Doppelhofer (1994), highlighted that intertemporal budget constraints have a substitution and an income effect on individual household behavior. This means that when the economy experiences an increase in the interest rates, the substitution effect will lead to more consumption in the future and less in the present by an individual who is more inclined to save. On the other hand, the income effects on an individual who is a borrower will be negative and for a lender will be positive.

### 3.2. Importance of Ricardian Equivalence

There is similarity between Ricardian theory and the permanent income lifecycle household theory, but the former includes government taxes, debt and expenditure on purchases in its computations (Malley and Molana 2002). The Ricardian equivalence hypothesis attributes to David Ricardo (1772–1823) views that tax collection and public borrowing establish equivalent types of financing public expenditure. The justification behind this view is that the government is expected at some future time to reclaim its debt (Kotlikoff 2018). Barro's treatment of Ricardian comparability has been tended to principally in the setting of national government finance.

The study of Ahiakpor 2013 focused upon the modern Ricardian equivalence and thereby probably abstained from examining whether increased government spending harms an economy's development. To that end, Hayo and Neumeier (2017) pointed out that the Ricardian equivalence theorem (RET) suggests that fiscal stimuli that can be explained in terms of deficit-financed public spending hikes or tax cuts would lead to a crowding out of private consumption, thereby decreasing the effectiveness of fiscal policy in boosting economic activity. That stipulates that a person's consumption is determined by the lifetime present value of his after-tax income. Thus, the underlying idea behind Ricardo's theory is that no matter how a government chooses to increase spending, whether with debt financing or tax financing, the outcome is the same and demand remains unchanged. In the similar context, the study of (Banzhaf and Oates 2013) pointed out the economist's general affirmation that consumers ought to be indifferent between bond financed and tax financed investments, or what is called "debt illusion".

The study of Bernheim and Ricardian (1987) focused upon the household behavior with regard to change in the taxes. The study points out that a present tax cut will not increase consumption since non-myopic people will see this arrangement as an increase of taxes later. However, these results are considered under the assumptions of an infinite horizon, non-distortionary taxes, an absence of liquidity constraints, farsighted people, and an ideal capital market. In this regard, the study of Nadenichek (2016) contends that Ricardian equivalence offers a theoretical benchmark, providing an initial point to break down the impacts of government debt on the economy. Ricardian equivalence theory can still be used to explain phenomena such as the case of decreased savings in the US economy in the last twenty years, and the European experience of expansionary fiscal contracts in the 90s. The visibility of Ricardian theory is also seen in the concept of debt neutrality which happens when the government is able to balance the amount of taxes received and the spending, which is financed by public debt and is said to have no effects on real variables and does not influence aggregate demand.

### 3.3. Relevance of Ricardian Equivalence in Turkey

The paradigm shift in Turkish economy after 1980 has made gradual change in the ratio of budget items, including both internal and external debts, grow rapidly. Increase in public debt in turn created high interest rates in the country, which attracted short term capital flow from abroad.

Orhan and Nergiz (2014) studied the scenario of debt in Turkey and found out that generously high current account and budget deficits are regularly accused as the fundamental sources of macroeconomic instability which led to the two financial crises during the post-1990 period. In light of this, the study of (Odim et al. 2014) highlighted that Turkey has turned out to be reliant on the developments of "interest rates", "exchange rates", and "inflation rates" because of the borrowing and financing requirements. In the similar context, Darrat (2006) in his study, states that the country's financial market is developing but is so far very sensitive. Credit unavailability leads to differential rates in borrowing. Fiscal spending helps the private investments flow, but private and public investments are still needed. There is also a need for expansionary fiscal policies so that public and private sectors can complement each other. After the financial liberalization endeavors of the 1980s, budget deficits have been financed exclusively by new debt creation and the mode of the domestic debt finance has often been through the domestic banking system. Additionally, the economy faced a high rate of volatility in inflation and the capital flows. The study by Boratav et al. (2007) revealed that the Turkish economy post 1980 went through major economic reforms which led to the fall in wage rates of labor. This fall in the wages was attributed to new capital-intensive techniques being introduced in the economy with the subsequent rise in foreign direct investment and trade balance. Due to the fall in the wages, the amount of consumer spending also declined. However, there was no impact on the savings ratio. The failure of the Turkish economy to sustain generally modest levels of CAD can be explained by the main primary source, public sector deficits. The recent development of the budget and current account deficiencies in Turkey has all the earmarks of being comparative with those seen in the U.S. Eichengreen et al. (2013) pointed out that, in any case, one ought to be incredibly careful in looking at the sustainability dynamics and adjustment mechanisms of the current account deficits in the US and a developing nation like Turkey,

as developing nations regularly experience the ill effects of "original sin" of the international financial system. That is, developing nations cannot borrow their own currencies from international financial markets. Foreign debt and borrowing in foreign exchange even domestically made Turkey vulnerable to the results of "original sin," as its revenues have been basically in domestic currency. Ayşe and Levent (2016) through their study, claimed that there is no direct clarification regarding the presence of Ricardian equivalence in Turkey based on both studies conducted within the country and outside it, but there are slight hints that point to its presence.

### 3.4. Criticisms of Ricardian Equivalence

Sachsida et al. (2011) pointed out that the Ricardian model has been criticized with the claim that the change in fiscal policy will not have an impact on any other agents related to it; however, Lucas pointed out the changes in behavior of agents with changes in fiscal policy. In this regard, Kotlikoff (2018) highlighted that, even though it is about impossible for Ricardian equivalence to hold precisely, the equivalence may depict the world approximately. The prevalence of Ricardian equivalence has been examined by loosening up its restrictive assumptions, such as capital market flaws, uncertainty, myopia, and distortionary tax collection. Under its relaxed assumptions, public debt may affect the economy.

Dalen (2011) in his study pointed out that the assumptions made by Ricardo in the theory have been highly doubted, for instance, in the case of neutrality of public debt wherein the concepts of dynasty and immigration play important roles. Even consumer heterogeneity and distribution of resources determine the debt neutrality issue. A standard criticism of the Ricardian equivalence hypothesis is defined through real-life circumstances which are described by vulnerability with respect to future income and furthermore tax risk, which keeps people from acting in accordance with normal desires. However, Ricardian equivalence does not hold in situations where the growth rate of the economy surpasses the rate of interest.

### 3.5. Empirical Evidence

A review of the literature highlighted secondary studies on the Ricardian equivalence theorem and its existence in Turkey. In this context, the current section provides an empirical review highlighting the aim, methodology, and findings derived by the previous studies as it is shown in Table 1.

**Table 1.** Empirical review.

| Author, Year | Aim of The Study | Methodology | Findings |
| --- | --- | --- | --- |
| (Ayşe and Levent 2016) | Testing the validity of twin deficit hypothesis for 2001–2014 | Toda Yamamoto test is used to examine the validity of the twin deficit hypothesis. Causality relation is derived between government's budget deficit, foreign trade deficit, exchange rate and interest rate. | Unidirectional causality is found in the study which does not support the Keynesian general Theory. On the other hand, bidirectional relationship is found between budget deficit and interest rates. This eventually impacts the level of consumer spending in the economy. |
| (Varol Iyidogan 2013) | Examining the validity of twin deficit relation in Turkey. | Budget Account and current account balance relationship has been analysed using descriptive statistics of the time series, stationarity of the series has been analysed using Zivot- Andrews test and lastly Causality test has been applied. | Current account balance and budget account balance are negatively related. |

**Table 1.** *Cont.*

| Author, Year | Aim of The Study | Methodology | Findings |
|---|---|---|---|
| (Bilgili 1997) | To test the Ricardian Equivalence Theorem of income, taxes, debt (or deficit) on the level of private consumption. | Dickey–Fuller's unit root tests is used first followed by VAR forecast method and finally the study uses co integration test by Johansen. | The variables are related to each other, but this relation is not constant. No long-term relationship could be found between the effects of income, taxes and deficit on the level of private consumption. |
| (Ricciuti 2001) | Testing the role of stochastic models and intertemporal government budget deficit in proving Ricardian Equivalence theorem. | Permanent Income Hypothesis and Eulers equation test | The theorem is the most contested one and studies concentrating on debt neutrality may find the prevalence of the theorem |
| (Darrat 2006) | Examining the intertemporal relationship between government spending and taxation in Turkey | Stationarity tests using ADF, PP AND WS unit root procedure Johansen cointegration technique is applied to variables Granger Causality test is used on variables. | Long-run equilibrium relationship between government spending and taxation is found in Turkey which eventually impacted the govt spending and the taxes Increase in taxes would curb the deficit in current account. |
| (Odim et al. 2014) | To examine the relation between government deficits and interest rate, in Nigeria | VAR analysis and VECM test has been used | Government deficit and interest rate were found to be related in the short-run. |

## 4. Methodological Framework

### 4.1. Variables for the Study

In accordance with the aim of the study, the following variables were taken into consideration for Turkey:

**Domestic government borrowings (GD):** The term is used for a country's governmental debt securities that are issued to raise money, both by the center and the state. These securities act as the main source for the government to finance their rising expenses (Adofu and Abula 2010). The data on the variable were collected in terms of government debt in Turkey, in percent of GDP terms.

**Gross domestic savings (GDS):** Gross domestic savings is used as a proxy for private savings which is defined as the amount of money saved by a given household. The level of private savings is impacted by the factors such as the demographics, financial status, political instability in the country affecting individual decisions, macroeconomic uncertainty, personal income, and internal government policies (Gök 2014). For the current study, the data for the variable were collected in the form of gross domestic savings in Turkey in USD.

**Governmental final consumption expenditure:** The total amount of national income which is used for individual consumption such as education, housing, and healthcare. Apart from this, it includes the consumption required for collective consumption, including defense, justice, and so on. This category of expenditure includes the social transfers which are undertaken from the government to the households (Eurostat and OECD 2012).

**Household final consumption expenditure:** This expenditure comprises of all the purchases which are made by the resident households and the everyday needs, including food, clothing, transport, and durable goods. Apart from this, it includes the all the imputed expenditures including the agricultural products and owner-occupied imputed rents. Thus, all the goods and services which are brought by households in order to meet the everyday needs. The household's individual consumption

comprises general government expenditure and expenditure which directly benefits all the households (Emeka and Kelvin 2016).

*4.2. Data Collection Procedure*

The data for the variables in quantitative terms were collected from online sources. The data for private savings and household final consumption expenditure were collected from the World Bank (https://data.worldbank.org/) and the data for domestic government borrowings and government final consumption expenditure were acquired from https://www.imf.org/en/Data. In order to determine the long-term relationship between government domestic borrowings and private savings in Turkey, the time period that was used for the study was from 1980–2017. This time period was specifically chosen to examine the relevance of Ricardian equivalence after the phase of structural change and reforms which the Turkish economy underwent in 1980.

*4.3. Data Analysis Procedure*

The augmented Dickey–Fuller unit root test was employed in the analysis of the study to find out about the stationarity of the domestic savings and domestic government debt and find out whether there is any convergence between them (Lee and Chang 2008). Further, the study employed Johansen's cointegration test, which determined whether the null hypothesis of no cointegration was accepted or rejected in the study. The test also helped determine the cointegrating vectors in variables, and we expected there to be at least one (Katircioglu 2009). Following this, a VAR model was used to determine the time dependence of normal variables and their interdependence (Soares 2011). The VECM test and Granger causality were used at last to see whether the past and present values of domestic government debt help predict domestic private savings in the Turkish economy.

The data collected through online sources were entered in MS-Excel and were converted into logarithmic form to remove variability and present it as close to a normally distributed form as possible. The data were then imported to STATA for the further analysis. STATA was used to run the augmented Dickey–Fuller unit root test for stationarity, Johansen's cointegration test which was used to cointegrate the vector of variables, and the vector error correction model (VECM).

In order to evaluate the validity of the Ricardian equivalence theory, the validity of Ricardian equivalence theory was tested using the work of Ramsey (1928). The model equation for the model is specified as follows:

$$HFC = B_0 + B_1 GDS + B_3 GD + B_4 GFCE$$

where HFC is the household fixed consumption, GDS stands for gross domestic savings, GD is the government debt, and the GFCE is the gross final consumption expenditure. The relationship between household final consumption, gross domestic savings, and government debt should offer answers as to the extent of dependence of each variable on the other. The lags of these variables were to be useful in explaining how the consumption pattern of the households changes when the level of government debt changes. This was also meant to depict the extent of the relationship between the gross savings and the household final consumption expenditure. In order to validate the no debt policy of the Ricardian equivalence hypothesis, the debt variable must have a negative coefficient. The Ricardian equivalence stands correct if the coefficient of the government spending is less than zero. Primarily, the augmented Dickey–Fuller unit root test was conducted in order to check for the order of integration among the different variables. Further, to determine the long-run relationships among the variables; the autoregressive distributed lag co-integration technique was used with a different order of integration. The auto-regressive distributed lag co-integration technique is preferable for dealing with the integration techniques of order 0, 1 or the combination of both. In addition to this, this integration technique was thought to be useful, as the number of observations was only 38. However, this technique should be useful in the presence of an integrated stochastic trend of second order of

integration. Specifically, in this case, the ARDL technique offers a unified framework for estimating the relations of co-integration within the context of a single equation (Emeka and Kelvin 2016).

## 5. Empirical Analysis

### 5.1. Augmented Dickey–Fuller Unit Root Test

The test was used in the study to check for the stationarity of variables so that further tests could be applied to prove the relationship between the two. P Values for this test is shown in Table 2. It was found that the variables were nonstationary in their original form and so the results for their Dickey–Fuller test, shown by Tables A3 and A4, were non-negative and non-significant. The p-value for the variable in the natural form was found to be 0.6742, which is much higher than the critical values and the significant value of 0.05 at 5% significance. Therefore, for the variable "GDS" (gross domestic savings), the null hypothesis could not be rejected; the variable was not stationary. Further, the variable "GD" (government debt) was also found to be non-stationary in its natural form and the null hypothesis could not be rejected because the p-value was found to be 0.2723, which is much higher than the critical values and significant value at 5% significance. Apart from this, the household final consumption and gross final consumption expenditure were also non-stationary with the p-value of 0.6306 for this period.

**Table 2.** Results of the augmented Dickey–Fuller.

| Variables | *p*-Value |
|:---:|:---:|
| GDS | 0.6742 |
| GD | 0.2723 |
| HFC | 0.6306 |
| GFCE | 0.7060 |
| logGDS | 0.1857 |
| logGD | 0.9963 |
| logHFC | 0.3697 |
| logGFCE | 0.3309 |
| dlogGDS | 0.0000 |
| dlogGD | 0.0000 |
| dlogHFC | 0.0000 |
| dlogHFC | 0.0000 |

The study of Luetkepohl and Xu (2009) pointed out to the importance of log transformations of each variable. Taking the log transformations of variables helps the study in getting optimal results. The variables by the log transformation are made more homogeneous. Therefore, the variables in the current study were also transformed into their log form to apply the augmented Dickey–Fuller unit root test and find stationarity in the log forms of variables, GDS, GD, HFC, and GFCE. The results for the test are depicted in Tables A3 and A4 (Appendix A). In this context, shown below are the p-values of the variables GDS GD, HFC, and GFCE in their real forms, log forms, and first differences of the log forms. These results were taken from the results in Tables A3 and A4 in Appendix A to point out to the series the current study lies under. After only the second differentiation, the *p*-values of the four variables under consideration became significant with the p-value of 0.000, indicating that after the second differentiation in the logarithmic form, the variables became stationary.

The *p*-value of unit root test for the log transformation of GDS, depicted by logGDS, was found to be 0.3415, which is again more than the significant value of 0.05 at 5% significance. Hence, the null hypothesis claiming the non-stationary of the log transformation of GDS could not be rejected. Therefore, the variable GDS was found to be non-stationary even in its log transformation form. The second variable in the study, i.e., GD, was also found to be non-stationary in its log transformation form. The *p*-value for the logGD unit root test was found to be 0.0562, which is higher than the significant level of 0.05 at 5% significance. Hence, the null hypothesis of non-stationarity could not be rejected.

In this context, the study of Selcuk and Ertugrul (2001) pointed out that Turkey during the period of 1980–2017 faced a lot of imbalance in the economy, with inflation rates shooting up and the growth in the economy going down; the country fell into a crisis situation. Therefore, the variables such as GD, HFC, GDS, and GFCE seem to show non-stationarity. To fix this issue further in the study, the first difference of the log transformation form was taken and unit root test was applied to see the results.

Since the two variables, i.e., GDS and GD, were not found to be stationary even in their log forms, the augmented Dickey–Fuller unit root test was further run on the log forms of these variables. The results for the unit root test for the first differenced variables are depicted in the Tables A3 and A4 in Appendix A. The p-values for the first differences of log GDS, log GD, log HFC, and log GFCF were also non-stationary and were greater than 0.05, indicating non-stationarity among the variables. The differentiation to the second-degree level made the p-value significant, indicating that the four variables became stationary.

The $p$-value for the first difference of logGDS, depicted as dlogGDS, was found to be 0, which is less than the significant value of 0.05 at 5% significance. Therefore, the null hypothesis of non-stationarity in this case can be rejected. Hence, it can be said the variable GDS is stationary in its first difference of log transformation form. The same can also be said for the variable GD, whose p-value for the first difference of log transformation form, depicted as dlogGD, was found to be 0, and hence the null hypothesis was of non-stationarity could be rejected. Those two variables in the study were found to be stationary after their first difference; hence, they belong to the I (1) series which means these variables are integrated at first order.

*5.2. Johansen's Cointegration Test*

The study of Naidu et al. (2017) explained that the Johansen cointegration test is used with I (1) series of stationarity. The test was used to determine the cointegration between dependent and independent variables in the study. It helps define the number of relationships the variables have with one another. The study of Drakos (2001) pointed out that the Ricardian equivalence theorem could be proven by establishing a long-term relationship between variables. For this, Johansen's cointegration method was used to examine that whether there existed any relationship between the variables in the study. Therefore, the researcher used the test to find out the number of relationships that existed in the study between the variables, which brought the study one step closer to proving the existence of Ricardian equivalence in that period. Johansen's cointegration test uses the maximum rank limit to showcase the number of cointegrating relationships among variables in the study (Afzal 2012). The hypothesis is set for each rank, and the maximum rank for the cointegration result must be at least 1. The cointegration rank can be achieved by looking at the test statistic and the maximum statistic which should be greater than the critical value at the level of significance set in the study.

The results in Appendix A suggest that the null hypothesis stating that there is no cointegration between the variables rank 1 is incorrect. Therefore, the results point out that there is cointegration between variables in the study. This can be verified by looking at the t-statistic value, which was 25.3775, which is much higher than the critical value of 15.41 at 5% significance. The maximum statistic also confirms the same, since the value of the maximum statistic was 22.0371 which is much higher than the critical value of 14.07 at 5% significance.

The alternative hypothesis was accepted in this study since the value of the t-statistic, 3.3404, and that of the maximum statistic, were the same. The value is less than the critical value of 3.76 at 5% significance. There was no value at rank 2 that could be checked to claim the rejection or acceptance of the hypothesis $H_2$. Hence, the results of Johansen's cointegration test point to the presence of a maximum of 1 cointegrating equation between variables.

The results shown in Table A7 clearly indicate that at the maximum rank of 0, the trace value (25.3775) exceeds the critical value (15.41). Therefore, the null hypothesis of no cointegration was rejected in this case. Hence, there is cointegration between GDS and GD at maximum rank 0. Similarly, the maximum value (22.0371) exceeds the critical value (14.07). Therefore, the two variables are cointegrated.

Further, for maximum rank 1, the null hypothesis is that there is cointegration of one equation, and the alternative hypothesis is that there is no cointegration of one equation. At maximum rank, the trace statistic (3.3404) does not exceed the critical value (3.76). The same is the case with the maximum statistic. Therefore, the null hypothesis must be accepted in this case. Thus, the variables GDS and GD were cointegrated of one equation.

*5.3. Specifying the Static Model*

Before applying the ARDL regression analysis, it is very important to specify the static model, which must be the econometric regression equation. The distributed lag model taking into consideration the regressors is defined by the following equation:

$$y_t = \alpha + \beta(L)x_t + u_t = \alpha + \sum_{s=0}^{\infty} \beta_s x_{t-s} + u_t,$$

The above equation clearly specifies the error taking into consideration the regressors $x$ on $y$. The above process is the infinite moving average with the specified lag weights and the lag distribution. In such a situation, it becomes very important to specify the lag distribution effectively which becomes zero beyond attaining the q periods. Another way would be considering the average and declining lag weights with more lags while requiring a minimal number of parameters.

In all kinds of equation estimation, it is very important to specify the lag length prior to the estimation. There is no specified method under economic theory which gives any information about the length of the lag. Thus, it becomes important to choose an appropriate method for identifying the length of the lag. Additionally, an alternative would be to test the significance of the all terms when examining the marginal coefficients of the lag terms.

The dependent variables comprise of set of regressors wherein the goodness of fit is measured through the Akaike information criterion and Bayesian information criteria. With the given number of coefficients and the residuals, the information criterion for minimizing the sum of the squared residuals is to choose the model with the smallest values of AIC and SBIC. For the current model, the results of the AIC and HQIC as per the selection order criteria with lags equal to 0, 1, 2, 3, and 4 are given in Appendix A. The results clearly indicate that among all the values, the lag values of 0, 1, 2, 3, and 4, the minimum value is for the AIC and HQIC with the lag of 0. The table in Appendix A indicates that the AIC and HQIC values for the lags are −5.12671 and −5.06568, with HFC, GFCE, GD, and GDS having the lowest values.

*5.4. Auto-Regressive Distributed Lag Model*

The economic analyses which were used to determine the long-run relationships among the variables used the autoregressive distributed lag co-integration technique. The results shown in Appendix A highlight the results of the auto-regressive distributed lag models. The independent variables included the government final consumption expenditure, gross domestic savings, and government debt. On the other hand, the dependent variable was the household final consumption expenditure. The results shown in Appendix A clearly indicate that the *p-value* was significant for the government final consumption expenditure. With one unit of increase in the government final consumption expenditure, the household final consumption increased by 0.72022 units. The results of the ARDL regression analysis indicate cointegration depicts the long-run relationship between the household and the government expenditure. The results of the autoregressive distributed lag model are shown in Tables A3 and A4 of Appendix A.

In this context, the study conducted by (Nyambe and Kanyeumbo 2015) highlighted that government expenditure is the crucial stimulant with the economic activities. If the government offers better quality of public goods through better education, healthcare, and other services, then the households are still willing to spend greater amounts to improve the standard of living. In case of stable inflationary pressure, it would be appropriate to extend the government expenditure.

In order to determine whether there is the causality among variables in the long-run or short-run, the researcher used a vector error correction model in the study. The study of Asari et al. (2011) points out that the results of cointegration test define whether VECM is to be performed in the study. If the maximum rank in the cointegration test is 1 or greater, VECM is applied to the model to find out the long-run relationship that exists between variables. If the test of cointegration would have pointed to a maximum rank of 0, then the study could not have proved any long-run relationship between variables. In the study of Andrei and Andrei (2015) VECM was explained to take care of deviations and short-run changes in the equilibrium situation. The dependent variable in the VECM model was assumed to be endogenously produced and the independent variables were assumed to be exogenously produced so that long-run and short-run associations between the variables could be determined.

The results of VECM Model are shown by Table A7 in Appendix A. The term "ce1" represents the cointegrating equations in the model. The value of ce1 when GDS is the dependent variable and GD is the independent variable is $-1.332819$, which is negative, and the *p*-value is 0, which is below the critical value of 0.05 under 5% significance. Therefore, there exists a long-term causality between GDS and GD.

Furthermore, for short-term causality between GDS and GD, the value of individual lag coefficient of GD ($-0.1205058$) is a negative number, but the *p*-value (0.605) is not significant at a 5% critical level of 0.05. Thus, there exists no short-term causal relationship between GDS and GD.

The value of ce1 in a case of GD being the dependent variable and GDS being the independent variable is $-0.1220723$, which is negative, but the *p*-value, 0.518, is not significant at the 5% critical level. Hence, there exists no long-run causality between GD and GDS. The causality does not hold in the short-run as well because the value of individual lag coefficient is 0.0737549, which is positive, and the *p*-value 0.534 is not significant at the 5% critical level.

The term "ce1" represents the cointegrating equations in the model. The value of ce1 when GDS is the dependent variable and GD is the independent variable is $-1.332819$, which is negative, and the *p*-value is 0, which is below the critical value of 0.05 under 5% significance. Therefore, there exists long-term causality between GDS and GD.

## 5.5. Granger Causality Test

The study of Kumar Narayan and Smyth (2004), points out that cointegration exists in the study then causality will also exist, through the ECM model. In order to determine the direction of causality, granger causality test is applied in studies involving multivariate analysis. According to the study of (Foresti 2007), granger causality can be used for two variables with lags, more than two variables and lastly using VAR model so that simultaneity can be maintained in variables. Therefore, the study uses the third way of analyzing Granger causality among the variables. Causality could be obtained between variables in either a unidirectional way where only one variable is proved to be the cause of the other, bidirectional causal relationship wherein both the variables affect each other, or they could be independent of each other. The results of the granger causality test are shown in Table A6 in the Appendix A.

In case of granger causality, the null hypothesis is the lagged value of one variable doesn't granger cause another variable. The findings of the results indicate that the gross domestic saving granger causes government debt with the *p*-value of less than 0.05. Therefore, the null hypothesis is rejected. Additionally, the government debt granger causes all other variables with less than 5% level of significance. Further the household final consumption also granger causes the government debt. Lastly, the government final consumption expenditure granger causes the government debt with less than 5% level of significance. Therefore, the presence of Granger Causality is bi-directional. This means that the government debt impacts the private savings in the economy. On the other hand, the private savings impact the extent of government debt in the economy. Thus, it can be said that the Ricardian equivalence theory is applicable in this scenario which means that increase in the government borrowing will have considerable impact on the private savings in Turkey. Similarly, low level of savings cannot generate a high level of external debt for the economy.

## 6. Conclusions

The present study examined the relevance Ricardian Equivalence theory in Turkey. For this purpose, the researcher explored the long-run relationship between the government domestic borrowings and private savings, using time series data for the period 1980–2017. Primarily, the researcher used the Dickey–Fuller test to examine the stationarity among the variable. The results of the test highlighted that the variables become stationary after taking the logarithmic and differentiation form. In the empirical analysis, Johansen cointegration test was used to point out to the existence of a long-run relation between the variables, gross domestic savings (GDS), which is used as a proxy for private savings in Turkey and government debt (GD), which is a proxy for Government domestic borrowings in the country.

The results imply to the presence of Ricardian equivalence in Turkey. This also implies that individuals like to save more with increase in Government debt in the economy. The analysis further uses VECM, VAR and Granger causality tests to point to the existence of a bilateral relationship between the two variables which implies that the variables impact each other. The relationship between the two variables was also found out to be positive which means that with increase in government debt the gross domestic savings increase. This clearly indicates that Ricardian equivalence theory is applicable in the Turkish economy. The results are in consistency with the literature reviewed in the study. The Ricardian equivalence points out that the government debt in Turkey can be financed by private savings. This means any tax cuts in the present times would be compensated with a tax increase in the future and public is aware about this scenario. Hence, the public tend to save more during the period of significant tax cuts. The variables used in the study were strategically picked to test the theory of Ricardian Equivalence, however, the study uses time series data from 1980–2017.

Further, the estimated equation indicated that the least values are valid for the AIC and BIC models with the lag of zero. Considering zero lags within the variables, the ARDL tests results indicated that the household consumption would vary according to the government consumption expenditure considering the validation of the Ricardian equivalence theory. As the government increases their expenditure, the households would vary their consumption pattern over a period of time. The households would increase their savings in order to deal with the threat of higher taxes in the near future.

This implies that the study resembles captures the long term. Further, the present study only used two variables including the government borrowings and private savings to prove the existence of the Ricardian theory in Turkey. Multiple analysis can be done in further studies by taking more related variables and can prove if the theory still holds in those situations. The studies can use consumption in the economy as a variable and fiscal variable like tax revenue to examine the long-run relationship between these variables and the government debt variable. In this regard, the study of (Korkmaz 2015) highlighted that the Turkish economy was continuously witnessing limited availability of capital and low savings rate. A low level of savings rate has in turn raised the consumption expenditure in turn generating a high amount of external debt in the Turkish economy. Apart from this, the economic crisis that emerged in the early 2000 caused Turkey to borrow huge amounts of funds from the IMF. The high debt rate generated instability and lead to large volatility within the macroeconomic variables which in turn negatively impacted the economic growth scenario. Additionally, one of important findings of the study is that as the government increases its fixed expenditure, the household consumption increases to only proportion of rise in the government expenditure. This is consistent with the permanent income hypothesis as the rise in government expenditure could be a reason for rise in disposable income only for a definite period of time. The consumption will rise only due to permanent rise in income. Further, it also depicts the rational expectations and the altruism effect considering the needs of the future generation.

The researcher used the Johansen cointegration test after considering the time series properties of the series. Additionally, the error correction method was used by the researcher to test the extent of validity for the Ricardian equivalence theorem. Apart from this, the findings also indicate that a rise in the government debt is associated with the rise in the private consumption. This is due to the fact as government increases its expenditure, it has to finance it through borrowings. The results indicated that with rise in expenditure, the household consumption would also increase—depicting it with

consistency in the Ricardian equivalence. The households perceive that the debt of the government is part of wealth and eventually they increase their consumption, providing evidence for the existence of Ricardian equivalence. However, during the long-run, the increase in the government debt would result in a loss of wealth of the tax payers or the households. Due to this, these household will have to curtail down their consumption in long-run.

After the liberalization of Turkish economy in 1980's there was a sharp increase of public borrowing in financing the budget deficits as an alternative way of direct taxation. The main outcome was high interest rates due to crowding out effect and deindustrialization of country with rent seeking activities due to the high rate of return on treasury bills. When the share of service sector increased to USD 25,000 GDP per capita with a declining share of industry in the Western world, Turkey reached a point of a low level of GDP compared to developed countries. Thus, the economic development performance of Turkey has up and downs due to the insufficient savings.

**Funding:** This research received no external funding.

**Conflicts of Interest:** The authors declare no conflict of interest.

## Appendix A

**Table A1.** Dickey–Fuller Test.

| | Test Statistic | 1% Critical Value | 5% Critical Value | 10% Critical Value | | | |
|---|---|---|---|---|---|---|---|
| Z(*t*) | −1.787 | −4.27 | −3.552 | −3.211 | | | |
| MacKinnon approximate *p*-value for Z(*t*) = 0.7108 | | | | | | | |
| D.GDS | | Coef. | Std. Err. | *t* | *p* > *t* | [95% Conf. Interval] | |
| | GDS | | | | | | |
| | L1. | −0.1334065 | 0.074635 | −1.79 | 0.083 | −0.285084 | 0.018271 |
| | _trend | $1.15 \times 10^9$ | $5.20 \times 10^8$ | 2.2 | 0.034 | $8.94 \times 10^7$ | $2.20 \times 10^9$ |
| | _cons | $-4.92 \times 10^9$ | $5.51 \times 10^9$ | −0.89 | 0.378 | $-1.61 \times 10^{10}$ | $6.27 \times 10^9$ |
| | Test Statistic | 1% Critical Value | 5% Critical Value | 10% Critical Value | | | |
| Z(*t*) | −0.862 | −4.27 | −3.552 | −3.211 | | | |
| MacKinnon approximate *p*-value for Z(*t*) = 0.9600 | | | | | | | |
| D.GD | | Coef. | Std. Err. | *t* | *p* > *t* | [95% Conf. Interval] | |
| | GD | | | | | | |
| | L1. | −0.0584846 | 0.067825 | −0.86 | 0.395 | −0.1963222 | 0.079353 |
| | _trend | $4.84 \times 10^8$ | $7.00 \times 10^8$ | 0.69 | 0.494 | $-9.38 \times 10^8$ | $1.91 \times 10^9$ |
| | _cons | $4.71 \times 10^9$ | $6.33 \times 10^9$ | 0.74 | 0.462 | $-8.16 \times 10^9$ | $1.76 \times 10^{10}$ |
| | Test Statistic | 1% Critical Value | 5% Critical Value | 10% Critical Value | | | |
| Z(*t*) | 1.946 | −4.27 | −3.552 | −3.211 | | | |
| MacKinnon approximate *p*-value for Z(*t*) = 0.6306 | | | | | | | |
| D.HFC | | Coef. | Std. Err. | *t* | *p* > *t* | [95% Conf. Interval] | |
| | HFC | | | | | | |
| | L1. | −0.1561244 | 0.080237 | −1.95 | 0.06 | −0.3191865 | 0.006938 |
| | _trend | $2.89 \times 10^9$ | $1.40 \times 10^9$ | 2.07 | 0.047 | $4.62 \times 10^7$ | $5.73 \times 10^9$ |
| | _cons | $-7.07 \times 10^9$ | $1.29 \times 10^{10}$ | −0.55 | 0.587 | $-3.33 \times 10^{10}$ | $1.92 \times 10^{10}$ |
| | Test Statistic | 1% Critical Value | 5% Critical Value | 10% Critical Value | | | |
| Z(t) | −1.797 | −4.27 | −3.552 | −3.211 | | | |
| MacKinnon approximate *p*-value for Z(*t*) = 0.7060 | | | | | | | |
| D.GFCE | | Coef. | Std. Err. | *t* | *p* > *t* | [95% Conf. Interval] | |
| | GFCE | | | | | | |
| | L1. | −0.1040432 | 0.057884 | −1.8 | 0.081 | −0.2216769 | 0.013591 |
| | _trend | $5.47 \times 10^8$ | $2.44 \times 10^8$ | 2.24 | 0.031 | $5.15 \times 10^7$ | $1.04 \times 10^9$ |
| | _cons | $-2.31 \times 10^9$ | $2.59 \times 10^9$ | −0.89 | 0.379 | $-7.57 \times 10^9$ | $2.95 \times 10^9$ |

**Table A2.** Dickey–Fuller test.

|  | **Test Statistic** | **1% Critical Value** | **5% Critical Value** | **10% Critical Value** | |
|---|---|---|---|---|---|
| Z($t$) | −2.831 | −4.27 | −3.552 | −3.211 | |

| MacKinnon approximate *p*-value for Z($t$) = 0.1857 | | | | | |
|---|---|---|---|---|---|
| **D.logGDS** | **Coef.** | **Std. Err.** | ***t*** | ***p > t*** | **[95% Conf. Interval]** |
| logGDS | | | | | |
| L1. | −0.35413 | 0.125075 | −2.83 | 0.008 | −0.60831 | −0.09994 |
| _trend | 0.029548 | 0.010499 | 2.81 | 0.008 | 0.008212 | 0.050885 |
| _cons | 8.273331 | 2.905278 | 2.85 | 0.007 | 2.369096 | 14.17757 |

|  | **Test Statistic** | **1% Critical Value** | **5% Critical Value** | **10% Critical Value** | |
|---|---|---|---|---|---|
| Z($t$) | 0.314 | −4.27 | −3.552 | −3.211 | |

| MacKinnon approximate *p*-value for Z($t$) = 0.9963 | | | | | |
|---|---|---|---|---|---|
| **D.logGD** | **Coef.** | **Std. Err.** | ***t*** | ***p > t*** | **[95% Conf. Interval]** |
| logGD | | | | | |
| L1. | 0.025154 | 0.0802 | 0.31 | 0.756 | −0.13783 | 0.188139 |
| _trend | −0.00524 | 0.007668 | −0.68 | 0.499 | −0.02082 | 0.010343 |
| _cons | −0.46224 | 1.877338 | −0.25 | 0.807 | −4.27745 | 3.352971 |

|  | **Test Statistic** | **1% Critical Value** | **5% Critical Value** | **10% Critical Value** | |
|---|---|---|---|---|---|
| Z(t) | −2.419 | −4.27 | −3.552 | −3.211 | |

| MacKinnon approximate *p*-value for Z($t$) = 0.3697 | | | | | |
|---|---|---|---|---|---|
| **D.logHFC** | **Coef.** | **Std. Err.** | ***t*** | ***p > t*** | **[95% Conf. Interval]** |
| logHFC | | | | | |
| L1. | −0.33162 | 0.137099 | −2.42 | 0.021 | −0.61024 | −0.053 |
| _trend | 0.027534 | 0.011921 | 2.31 | 0.027 | 0.003307 | 0.051761 |
| _cons | 8.083918 | 3.309086 | 2.44 | 0.02 | 1.359047 | 14.80879 |

|  | **Test Statistic** | **1% Critical Value** | **5% Critical Value** | **10% Critical Value** | |
|---|---|---|---|---|---|
| Z($t$) | −2.494 | −4.27 | −3.552 | −3.211 | |

| MacKinnon approximate *p*-value for Z($t$) = 0.3309 | | | | | |
|---|---|---|---|---|---|
| **D.logGFCE** | **Coef.** | **Std. Err.** | ***t*** | ***p > t*** | **[95% Conf. Interval]** |
| logGFCE | | | | | |
| L1. | −0.26642 | 0.106823 | −2.49 | 0.018 | −0.48352 | −0.04933 |
| _trend | 0.027595 | 0.011051 | 2.5 | 0.018 | 0.005137 | 0.050054 |
| _cons | 5.956746 | 2.364802 | 2.52 | 0.017 | 1.150891 | 10.7626 |

**Table A3.** (**a**) Dickey–Fuller Tests. (**b**) Auto-regressive distributed lag model.

(**a**)

|  | Test Statistic | 1% Critical Value | 5% Critical Value | 10% Critical Value | | |
|---|---|---|---|---|---|---|
| Z($t$) | −7.522 | −4.279 | −3.556 | −3.214 | | |

| MacKinnon approximate $p$-value for Z($t$) = 0.0000 | | | | | | |
|---|---|---|---|---|---|---|
| **D.dlogGDS** | **Coef.** | **Std. Err.** | *t* | *p* > *t* | **[95% Conf. Interval]** | |
| dlogGDS | | | | | | |
| L1. | −1.2599 | 0.167506 | −7.52 | 0 | −1.6007 | −0.91911 |
| _trend | 0.001751 | 0.003218 | 0.54 | 0.59 | −0.0048 | 0.008298 |
| _cons | 0.050001 | 0.068583 | 0.73 | 0.471 | −0.08953 | 0.189534 |

|  | Test Statistic | 1% Critical Value | 5% Critical Value | 10% Critical Value | | |
|---|---|---|---|---|---|---|
| Z($t$) | −5.721 | −4.279 | −3.556 | −3.214 | | |

| MacKinnon approximate $p$-value Z($t$) = 0.0000 | | | | | | |
|---|---|---|---|---|---|---|
| **D.dlogGD** | **Coef.** | **Std. Err.** | *t* | *p* > *t* | **[95% Conf. Interval]** | |
| dlogGD | | | | | | |
| L1. | −0.99783 | 0.174411 | −5.72 | 0 | −1.35267 | −0.64299 |
| _trend | −0.00315 | 0.001825 | −1.72 | 0.094 | −0.00686 | 0.000567 |
| _cons | 0.129583 | 0.043177 | 3 | 0.005 | 0.041739 | 0.217428 |

|  | Test Statistics | 1% Critical Value | 5% Critical Value | 10% Critical Value | | |
|---|---|---|---|---|---|---|
| Z(t) | −6.321 | −4.279 | −3.556 | −3.214 | | |

| MacKinnon approximate $p$-alue for Z($t$) = 0.0000 | | | | | | |
|---|---|---|---|---|---|---|
| **D.dlogHFC** | **Coef.** | **Std. Err.** | *t* | *p* > *t* | **[95% Conf. Interval]** | |
| dlogHFC | | | | | | |
| L1. | −1.09545 | 0.17329 | −6.32 | 0 | −1.44801 | −0.74288 |
| _trend | −0.00114 | 0.002696 | −0.42 | 0.676 | −0.00662 | 0.004349 |
| _cons | 0.09661 | 0.058684 | 1.65 | 0.109 | −0.02278 | 0.216003 |

(**b**)

| lag | LL | LR | df | *p* | FPE | AIC | HQIC | SBIC |
|---|---|---|---|---|---|---|---|---|
| 0 | 88.5908 | | | | $7 \times 10^{-8}$ | −5.12671 | −5.06568 | 4.94532 |
| 1 | 100.08 | 24.418 | 16 | 0.081 | $8.90 \times 10^{-8}$ | −4.89695 | −4.59178 | −3.98997 |
| 2 | 113.553 | 25.507 | 16 | 0.061 | $1.10 \times 10^{-7}$ | −4.70019 | −4.15088 | −3.06764 |
| 3 | 125.121 | 23.135 | 16 | 0.11 | $1.70 \times 10^{-7}$ | −4.43157 | −3.63813 | −2.07343 |
| 4 | 143.473 | 36.706 | 16 | 0.002 | $1.90 \times 10^{-7}$ | −4.57419 | −3.53661 | −1.49047 |

**Table A4.** Auto-regressive distributed lag model.

| **dlogHFC** | **Coef.** | **Std. Err.** | *z* | *p* > *z* | **[95% Conf. Interval]** | |
|---|---|---|---|---|---|---|
| dlogGFCE | 0.72022 | 0.08684 | 8.29 | 0 | 0.550017 | 0.890424 |
| dlogGD | 0.168922 | 0.116688 | 1.45 | 0.148 | −0.05978 | 0.397626 |
| dlogGDS | 0.140744 | 0.07342 | 1.92 | 0.055 | −0.00316 | 0.284645 |
| _cons | −0.00785 | 0.015577 | −0.5 | 0.614 | −0.03838 | 0.022678 |

**Table A5.** Vector auto-regression.

|  |  | Coef. | Std. Err. | *z* | *p* > *z* | [95% Conf. Interval] | |
|---|---|---|---|---|---|---|---|
| GDS |  |  |  |  |  |  |  |
|  | GDS L1. | 0.527108 | 0.225943 | 2.33 | 0.02 | 0.084268 | 0.969948 |
|  | GD L1. | 0.166852 | 0.06545 | 2.55 | 0.011 | 0.038572 | 0.295132 |
|  | HFC L1. | −0.0451 | 0.145242 | −0.31 | 0.756 | −0.32977 | 0.239573 |
|  | GFCE L1. | 0.616991 | 0.508445 | 1.21 | 0.225 | −0.37954 | 1.613524 |
|  | _cons | $3.80 \times 10^9$ | $4.55 \times 10^9$ | 0.840 | 0.404 | 0.456789 | $1.27 \times 10^{10}$ |
| GD |  |  |  |  |  |  |  |
|  | GDS L1. | −0.31434 | 0.233063 | −1.35 | 0.177 | −0.77113 | 0.142459 |
|  | GD L1. | 1.190752 | 0.067513 | 17.64 | 0 | 1.05843 | 1.323074 |
|  | HFC L1. | 0.024916 | 0.149819 | 0.17 | 0.868 | −0.26872 | 0.318557 |
|  | GFCE L1. | −0.12473 | 0.524469 | −0.24 | 0.812 | −1.15267 | 0.903213 |
|  | _cons | $7.16 \times 10^9$ | $4.70 \times 10^9$ | 1.53 | 0.127 |  | $1.64 \times 10^{10}$ |
| HFC |  |  |  |  |  |  |  |
|  | GDS L1. | −0.6244 | 0.481732 | −1.3 | 0.195 | −1.56857 | 0.319781 |
|  | GD L1. | 0.566603 | 0.139546 | 4.06 | 0 | 0.293098 | 0.840107 |
|  | HFC L1. | 0.563406 | 0.30967 | 1.82 | 0.069 | −0.04354 | 1.170348 |
|  | GFCE L1. | 1.503606 | 1.084054 | 1.39 | 0.165 | −0.6211 | 3.628313 |
|  | _cons | $1.59 \times 10^{10}$ | $9.71 \times 10^9$ | 1.64 | 0.102 | 0.89789 | $3.49 \times 10^{10}$ |
| GFCE |  |  |  |  |  |  |  |
|  | GDS L1. | 0.030714 | 0.094602 | 0.32 | 0.745 | −0.1547 | 0.21613 |
|  | GD L1. | 0.108039 | 0.027404 | 3.94 | 0 | 0.054329 | 0.16175 |
|  | HFC L1. | 0.002123 | 0.060813 | 0.03 | 0.972 | −0.11707 | 0.121313 |
|  | GFCE L1. | 0.705917 | 0.212885 | 3.32 | 0.001 | 0.288671 | 1.123164 |
|  | _cons | 0.987896 | $1.91 \times 10^9$ | −0.26 | 0.794 | 0.56459 | $3.24 \times 10^9$ |

**Table A6.** Granger Causality.

| Equation | Excluded | chi2 | df | Prob > chi2 |
|---|---|---|---|---|
| GDS | GD | 6.499 | 1 | 0.011 |
| GDS | HFC | 0.0964 | 1 | 0.756 |
| GDS | GFCE | 1.4725 | 1 | 0.225 |
| GDS | ALL | 16.259 | 3 | 0.001 |
| GD | GDS | 1.819 | 1 | 0.177 |
| GD | HFC | 0.02766 | 1 | 0.868 |
| GD | GFCE | 0.05656 | 1 | 0.812 |
| GD | ALL | 18.307 | 3 | 0.000 |
| HFC | GDS | 1.68 | 1 | 0.195 |
| HFC | GD | 16.486 | 1 | 0.000 |
| HFC | GFCE | 1.9238 | 1 | 0.165 |
| HFC | ALL | 23.705 | 3 | 0.000 |
| GFCE | GDS | 0.10541 | 1 | 0.745 |
| GFCE | GD | 15.543 | 1 | 0.000 |
| GFCE | HFC | 0.00122 | 1 | 0.972 |
| GFCE | ALL | 21.37 | 3 | 0.000 |

**Table A7.** Cointegration test.

| Johansen Tests for Cointegration | | | | | | |
|---|---|---|---|---|---|---|
| **Trend: Constant** | | | **Number of obs = 36** | | | |
| **Sample: 1982–2017** | | | **Lags = 2** | | | |
| | | 5% | | | | |
| | maximum | | trace | | critical | |
| rank | parms | LL | eigenvalue | statistic | | value |
| 0 | 6 | −1785.9538 | | 25.3775 | | 15.41 |
| 1 | 9 | −1774.9352 | 0.45781 | 3.3404 | | 3.76 |
| 2 | 10 | −1773.265 | 0.08861 | | | |
| | | 5% | | | | |
| | maximum | | max | | critical | |
| rank | parms | LL | eigenvalue | statistic | | value |
| 0 | 6 | −1785.9538 | | 22.0371 | | 14.07 |
| 1 | 9 | −1774.9352 | 0.45781 | 3.3404 | | 3.76 |
| 2 | 10 | −1773.265 | 0.08861 | | | |
| **Vector error-correction model** | | | | | | |
| **Sample: 1983–2017** | | | **No. of obs = 35** | | | |
| | | | **AIC = −1.246467** | | | |
| **Log likelihood = 30.81317** | | | **HQIC = −1.108405** | | | |
| **Det (Sigma_ml) = 0.0005893** | | | **SBIC = −0.8465201** | | | |
| Equation | Parms | RMSE | R-sq | chi2 | *p* > chi2 | |
| D_dlogGDS | 4 | 0.203618 | 0.6350 | 53.94156 | 0.0000 | |
| D_dlogGD | 4 | 0.135205 | 0.2910 | 12.72565 | 0.0127 | |

**Table A7.** *Cont.*

| Vector error-correction model | | | | | | |
|---|---|---|---|---|---|---|
| | Coef. | Std. Err. | z | p > |z| | [95% Conf. Interval] | |
| D_dlogGDS _ce1 | | | | | | |
| L1. dlogGDS | −1.332819 | 0.2847097 | −4.68 | 0.000 | −1.89084 | −0.7747987 |
| LD. dlogGD | 0.0550099 | 0.1785852 | 0.31 | 0.758 | −0.2950107 | 0.4050306 |
| LD. | −0.1205058 | 0.2328727 | −0.52 | 0.605 | −0.5769279 | 0.3359164 |
| _cons | 0.0003087 | 0.0344236 | 0.01 | 0.993 | −0.0671603 | 0.0677777 |
| D_dlogGD _ce1 | | | | | | |
| L1. dlogGDS | −0.1220723 | 0.1890521 | −0.65 | 0.518 | −0.4926075 | 0.248463 |
| LD. dlogGD | 0.0737549 | 0.1185836 | 0.62 | 0.534 | −0.1586648 | 0.3061745 |
| LD. | .5507026 | 0.1546314 | −3.56 | 0.000 | −0.8537746 | −0.2476305 |
| _cons | −0.0033707 | 0.0228578 | −0.15 | 0.883 | −0.0481712 | 0.0414298 |

| Cointegrating equations | | | |
|---|---|---|---|
| Equation | Parms | chi2 | p > chi2 |
| _ce1 | 1 | 0.4366088 | 0.5088 |

| Identification: beta is exactly identified | | | | | | |
|---|---|---|---|---|---|---|
| **Johansen normalization restriction imposed** | | | | | | |
| beta _ce1 | Coef. | Std. Err. | z | p > |z| | [95% Conf. Interval] | |
| dlogGDS | 1 | | | | | |
| dlogGD | −0.2133542 | 0.3228902 | −0.66 | 0.509 | −0.8462074 | 0.4194991 |
| _cons | −0.0523882 | | | | | |

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
