# Peer review of "Testing the Ricardian Equivalence Theorem: Time Series Evidence from Turkey"

_economies, doi:10.3390/economies8030069_

Round 1

Reviewer 1 Report

The article raises an important macroeconomic issue, namely postulating the Ricardian equivalence theorem in the Turkish economy. He states that financing government expenditure through the sale of government bonds (borrowing by the government) is equivalent to covering them by raising taxes. The issue of fiscal policy effectiveness here basically boils down to whether government bonds are treated by households as part of their net assets. Assuming that this is indeed the case, households will reduce their savings at the expense of other forms of hoarding and consumer spending will increase. Only in this case will fiscal policy prove effective. However, this means that the issue of future tax liabilities is ignored.

This article attempts to verify the Ricardian equivalence based on empirical data from the Turkish economy. My comments are as follows:

  1. It is necessary to think carefully about the results of empirical research presented in the article and the conclusions that can be formulated on their basis. It has been shown that there is a mutual relationship between the variables: gross domestic savings (GDS) and government debt (GD) that an increase in GDS causes an increase in GD and an increase in GD causes an increase in GDS (lines 390–395). In cybernetics, this means that there is positive feedback between these variables. It is a self-accelerating process that must sooner or later destabilize not only the financial sector, but also the real economy. This is not a demonstration of Ricardian equivalence theory. It can be some form of financialization. This is a serious problem that must be solved somehow. You may need to verify the calculation.
  2. There is insufficient evidence in the article for the Ricardian equivalence. Even if one-way relationship between GD and GDS were found, it would have to be demonstrated that the changes in these variables are equivalent in monetary terms (in real, absolute or relative terms). In addition, ceteris paribus, real national income would have to remain unchanged. It is difficult to prove. It is easier to show that the Ricardian equivalence does not occur.
  3. There are no well-formulated research hypotheses in the article (lines 84-87). The hypothesis in science is the researcher’s answer to the research question. It is expressed in the form of a relationship between a dependent variable and an independent variable. The article does not explicitly pose research questions, but one can guess from the context what they could be. Nevertheless, the hypotheses need to be reformulated.
  4. In the theoretical part, a synthetic statement of the assumptions underlying the Ricardian Equivalence Theorem (families act as infinitely lived dynasties because of intergenerational altruism, capital markets are perfect and so on) is needed.
  5. One way to save the article is to assume that if the calculations are correct, the result discussed in point 1 leads to a contradiction. And this means that the Ricardian equivalence does not occur in the Turkish economy.
  6. The economic interpretation of the results obtained should be strengthened, whatever they may be. How can the discovered relationships be used in economic policy? Do the economic effects of the identified dependency support the country’s economic growth and development, or are they rather a barrier to growth? Is fiscal policy effective?
  7. According to the above remarks, it is necessary to at least reformulate the title of the article, introduction, research aim & objectives, and conclusion. The easiest way is to follow the path presented in point 5 (if the calculations are correct).

In my opinion, the publication of the article is an open matter and depends on the answer to the above issues.

Reviewer 2 Report

The paper aims to test to Ricardian Equivalence for the Turkey economy. To this, the authors apply a cointegration analysis. The introduction and the literature review are well exposed. However, there are shortcomings in the research.

First, the econometric analysis is well done, but unfortunately, I do not think it is correct. To apply a cointegration analysis, we need more observations (at least 70 obs). 36 are few. Therefore I suggest to the authors to use quarterly data. If this is not possible, I recommend the authors to adopt an ARDL model that needs fewer observations.

Second, the methodology is not explained. There are no formulas about it. I strongly suggest to include it.

Third, the tables in the appendix are not clear. I suggest to modify them.

Fourthly, I don’t think that a bivariate relationship between Domestic Government Borrowings and Private Savings can capture the Ricardian equivalence. There are several other factors (variables) that the authors do not consider (see for example the old works of Ramsey (1928) and Park (1997); Permanent Income Hypothesis (PIH)). A possible solution would be that of estimating the aggregate consumption function.

Ramsey (1928) “A Mathematical Theory of Saving”, The Economic Journal, Vol. 38 Issue 152, December.

Park W.G. (1997), A Permanent Income Model of Public Consumption. Journal of Macroeconomics, Fall, Vol. 19, No. 4, pp. 753-769.

Round 2

Reviewer 1 Report

The corrected version of the article is not much better than the original one, but the Author only slightly followed my instructions. In fact, only the basic assumptions on which the Ricardian equivalence theorem is based have been added to the content. However, the most important comments were not taken into account, which means that the manuscript still suffers from serious defects. I will formulate my comments once more in points, and I will simply repeat some of them:

  1. The new version did not correctly formulate research hypotheses (lines 84-87). Once again, I remind you that the research hypothesis is an affirmative sentence about the relationships between variables. The answer to the research question proposed by the researcher is considered to be a hypothesis in science. It is expressed in the form of a relationship between a dependent variable and an independent variable. Then the hypothesis is confirmed or refuted. There is only one hypothesis at stake here:

In the Turkish economy, the Ricardian equivalence does not occur, as evidenced by positive feedback between Gross domestic savings (GDS) and Government Debt (GD).

  1. It has been shown that there is a mutual relationship between the variables Gross domestic savings (GDS) and Government Debt (GD) that an increase in GDS causes an increase in GD, and an increase in GD causes an increase in GDS. In cybernetics, this means that there is positive feedback between these variables. It is a self-accelerating process that must sooner or later destabilize not only the financial sector, but also the real economy. This is not a demonstration of Ricardian equivalence theory. This contradicts the basic thesis of the article. Cybernetics makes it possible to organize research not only in economics, but also in many other disciplines of science.
  2. There is insufficient evidence in the article for the Ricardian equivalence. Even if one-way relationship between GD and GDS were found, it would have to be demonstrated that the changes in these variables are equivalent in monetary terms (in real terms). In addition, ceteris paribus, real national income would have to remain unchanged. It is difficult to prove. It is easier to show that the Ricardian equivalence does not occur. The Ricardian equivalence cannot be demonstrated on the basis of two variables, because there is simply not enough data. In addition, the time series are too short. It can only be said that there is no evidence of the Ricardian equivalence in the empirical data collected on the Turkish economy.
  3. When demonstrating the veracity of the Ricardian equivalence, you need to know if government bonds are treated by households as part of their net assets.
  4. Once again I repeat: the article can be saved, but you need to completely change the interpretation of the results.

Author Response

All done

Reviewer 2 Report

Notes for Reviewer 3
1. There is shortage of data since there is no quarterly data on that. In
addition, it seems it is not so easy to apply another method since I have to
make certain revision in short period of time.

Review: I understand that quarterly data are not available, but a cointegration analysis with only 36 observations can lead to estimates that are not entirely accurate.

2.
3. Yes tables in appendix moved in electronic transfer and well organised
before publication.

4. Every research has its own limitations and the attempts to add much more variables would make much more complex to evaluate. Thus my intention to make is less complex.

Review: I disagree with the authors' answer (comment 4) "Every research has its own limitations and the attempts to add much more
variables would make much more complex to evaluate. " I do not think that adding other variables would make the evaluation more complicated; in my view, it would be a more complete evaluation.

Review: Unfortunately, I have to reject the paper, as none of my suggestions has been applied (data, methodology, formulas). The main limitation concerns the few data used to apply cointegration analysis.

Author Response

All of of the conclusions of reviewer done and article rearranged with model and new research methodologies. 

Round 3

Reviewer 1 Report

In the third version of the article I do not see any significant changes compared to the previous two versions. There is still no evidence of the Ricardian equivalence in the Turkish economy. The author did not address my questions or doubts at all. In reply, he wrote that cybernetics does not refer to the issue discussed in the article. This is a mistake. Cybernetics applies wherever there is a flow of matter, energy and information. The economy is a cybernetic system. Thanks to cybernetics you can avoid many mistakes in reasoning. The result obtained by the Author can be interpreted only in such a way that the discovered relationship between the government domestic borrowings (GDB) and private savings (PS) is not contradictory to Ricardian equivalence theorem.

The Ricardian equivalence has not been proven for the following reasons:

  1. The initial part of the article lists the assumptions (lines 99-107) that must be met in order for the equivalence theory to apply. These assumptions are not referenced in the conclusions. We do not know, therefore, whether they are met in the Turkish economy.
  2. Confirmation of the Ricardian equivalence requires that more variables be taken into account. Even if a relationship were found between GDB and PS, it should be demonstrated that changes in these variables are equivalent in monetary terms (in real terms). Only then can you talk about equivalence. In addition, ceteris paribus, real national income would have to remain unchanged. This is not in the article. Moreover, negative feedback must be demonstrated between real macroeconomic variables to demonstrate the inefficiency of fiscal policy.
  3. According to Ricardo, the problem of fiscal policy effectiveness comes down to whether government bonds are treated by households as part of their net assets. If this is the case, then households will reduce their savings at the expense of other forms of hoarding and consumer spending will increase. The author does not consider this problem at all.
  4. Time series consist only of 38 annual observations of selected macroeconomic variables. Of course, quarterly data would be better, but I understand that they may not be available. Annual macroeconomic data may contain unspecified amounts of random noise. Therefore, you need to be careful when making conclusions.

To sum up, the article requires significant changes in the evidence layer, which of course should be reflected in the economic interpretation of the results obtained. It has only been shown that the calculations are not contrary to the Ricardian equivalence. There is no evidence for other claims. In my opinion, maintaining the previous erroneous economic interpretation of calculations makes the article unsuitable for publication.

Author Response

I rearranged

Reviewer 2 Report

I really appreciate the author(s)’ effort. It was not easy to estimate a new model, enter more data and comment on it. Now the paper seems to be much more “correct” from an econometric point of view with the application of the ARDL model.

However, the paper still needs revision. In particular, I am referring to the citations of the tables and results. Unfortunately, there are some typos, and the article is a bit confusing.

For example:

Row 384 “The results shown in table 9 clearly” but in the appendix table 9 not exist, as well as in row 457 “The results of VECM Model are shown by Table 9”. There are two references for Table 9, but Table 9 not exist. Row 484 “The results of the VAR model are depicted by Table 9 in the appendix.”

I don’t’ understand where are the results of Vector Error Correction Model, well as the results of Johansen cointegration.

I strongly suggest to modify the tables; they are not clear. For example, Table 7: Vector auto-regression, there are many “#####” in the 95% Conf. Column.

I think with these changes, the paper is publishable. I really appreciate the effort made; it’s really close to the finish line.

Author Response

I did mentioned changes
